# State-of-the-Art on Advancements in Carbon–Phenolic and Carbon–Elastomeric Ablatives

**DOI:** 10.3390/polym16111461

**Published:** 2024-05-22

**Authors:** Amit Kumar, Chikesh Ranjan, Kaushik Kumar, M. Harinatha Reddy, B. Sridhar Babu, Jitendra Kumar Katiyar

**Affiliations:** 1Department of Mechanical Engineering, RTC Institute of Technology, Ranchi 835219, India; amit.mea.2103017@iiitbh.ac.in; 2Department of Mechanical Engineering, National Institute of Technology, Rourkela 769005, India; j.chikesh123@gmail.com; 3Department of Mechanical Engineering, Birla Institute of Technology, Ranchi 835215, India; kkumar@bitmesra.ac.in; 4Department of Mechanical Engineering, CVR Engineering College, Hyderabad 501510, India; harinathareddy.maddika@gmail.com; 5Department of Mechanical Engineering, Malla Reddy Engineering College, Hyderabad 500088, India; bsridhar8477@gmail.com; 6Centre for Research Impact and Outcome, Chitkara University, Chandigarh-Patiala National Highway (NH-7), Rajpura 140401, India

**Keywords:** linear ablation rate (LAR), mass ablation rate (MAR), ablative, phenolic resin, elastomeric resin, carbon, carbon and phenolic composites, carbon and elastomeric composites, pyrolysis, thermal resistance, thermal protection system (TPS)

## Abstract

Ablative composites serve as sacrificial materials, protecting underlying materials from high-temperature environments by endothermic reactions. These materials undergo various phenomena, including thermal degradation, pyrolysis, gas generation, char formation, erosion, gas flow, and different modes of heat transfer (such as conduction, convection, and radiation), all stemming from these endothermic reactions. These phenomena synergize to form a protective layer over the underlying materials. Carbon, with its superb mechanical properties and various available forms, is highlighted, alongside phenolics known for good adhesion and fabric ability and elastomers valued for flexibility and resilience. This study focuses on recent advancements in carbon-and-phenolic and carbon-and-elastomeric composites, considering factors such as erosion speed; high-temperature resistance; tensile, bending, and compressive strength; fiber–matrix interaction; and char formation. Various authors’ calculations regarding the percentage reduction in linear ablation rate (LAR) and mass ablation rate (MAR) are discussed. These analyses inform potential advancements in the field of carbon/phenolic and carbon/elastomeric ablative composites.

## 1. Introduction

The development of composite materials traces back to before the year zero, but scientific innovation in composites began in the early 20th century A.D [1,2]. In contrast, the early evolution of ablative composites began in the 1950s [3]. Ablatives are engineered composites designed specifically to withstand high-temperature environments [4,5]. When exposed to elevated temperatures, these composite materials gradually wear away, forming a charred surface. This charred surface acts as a protective shield, effectively guarding the underlying structure against intense heat and heat flow in situations involving elevated temperatures [6]. These composite materials are commonly utilized in materials demanding substantial thermal insulation, such as rocket nozzles, thermal barriers, heat shields, and vehicles re-entering the atmosphere [7,8,9,10].

The mechanism of ablation involves several interconnected processes, including thermal degradation, pyrolysis, gas generation, char formation, erosion, gas flow, and different modes of heat transfer [6]. Ablative composites primarily utilize carbon, aramid, ceramics, glass, etc., as reinforcements [11,12,13]. Some exceptional properties of carbon make it a widely used reinforcement in ablative composites; these properties include a notable strength-to-weight ratio, exceptional thermal stability, low thermal expansion, a high modulus of elasticity, exceptional chemical resistance, and commendable fatigue resistance, among others [14,15].

Organic polymeric resins (e.g., phenolics, elastomers, carbon, polyamides, epoxy resins), ceramics, etc., serve as matrices [16,17]. Phenolic resins often yield high carbon content during ablation, forming a significant amount of char residue on exposure to high temperatures. The presence of char serves as a protective barrier for the underlying structure, mitigating the erosion rate. Phenolic resins exhibit good adhesion to various reinforcements, such as fibers or particulates, enhancing the composite’s mechanical strength and load-transfer efficiency. They also demonstrate good thermal stability, maintaining structural integrity and mechanical properties under thermal loading conditions. Additionally, phenolic resins can be tailored to control mechanical properties and ablation behavior and exhibit good processability, imparting efficient fabricability to phenolic resins as ablative matrices. Generally, they offer good resistance to various chemicals, moisture, UV radiation, and other environmental factors [18,19].

Flexibility, resilience, and the ability to withstand high temperatures are common characteristics found in different elastomeric resins used as matrices for ablative composites. However, various elastomers possess unique properties, making them advantageous choices for ablative matrices [20,21].

The inherent properties of carbon, phenolic, and elastomeric resins make them well-suited for the fabrication of ablative composites. Recent innovation trends and advancements in ablative composites have extensively utilized carbon as reinforcement and phenolic and elastomeric resins as matrices. This article briefly discusses these trends and advancements in the following sections.

## 2. Carbon-Phenolic Ablative Composites

Carbon/phenolic ablative composites are utilized in aerospace applications due to their design flexibility, high yield of char formation, high thermal protection, light weight, etc. [9,10]. Carbon reinforcements are available in several forms, including continuous fiber, chopped fiber, fiber tow, fiber fabric, fiber prepreg, and particles. Continuous fibers are typically produced from precursor materials through a series of steps, including spinning, stabilization, carbonization, and sometimes graphitization. Chopped fibers are made by cutting the continuous thread into short lengths, and the production of fiber tows involves collecting and twisting individual carbon fibers to form a larger bundle. Carbon fiber fabric utilizes carbon fiber tows woven or knit into textile structures. Carbon fiber prepregs are produced by impregnating a resin matrix into continuous carbon fibers. Particulates are typically manufactured by milling or grinding the continuous fibers [22,23,24,25].

Phenolic resin is widely used in applications such as adhesives, coatings, and composites. It is produced from phenol and formaldehyde through chemical reactions, including condensation and polymerization.

Fillers are employed to hybridize carbon/phenolic ablative composites. The characteristics of filler-modified carbon-and-phenolic ablatives depend on factors such as filler particle size, filler inclusion percentage by weight, forms of carbon fibers, manufacturing techniques, etc. Various fillers, including silica, zirconium orthosilicate (ZrSiO_2_), silicon carbide (SiC), natural rubber latex, graphene nano-platelets, etc., have been used individually and in hybrid form to modify and enhance the characteristics of carbon/phenolic ablative composites. Silica inclusions have been found to improve microstructure, thermal stability, and anti-oxidation properties [26]. The adhesion between carbon fabric and phenolic resin was enhanced by the use of a ZrSiO_4_ sol coating [27]. Silicon oxycarbide filler contributed to increased heat-insulation capacity, elasticity, and compressive strength [28].

Hybridization of ablative composites has led to the optimization of several properties, including ablation rate, thermal stability, erosion resistance, flexural strength, tensile strength, shear strength, compression strength, fiber–matrix interaction, char production, and thermal conductivity. Among these properties, ablation rate is considered the most critical characteristic, measuring the thickness or mass loss per unit of time during pyrolysis. Linear ablation rate (LAR) quantifies thickness loss per unit of time, while mass ablation rate (MAR) estimates mass loss per unit of time. Table 1, provided below, presents the percentage decrease in both linear and mass ablation rates as documented across a range of research investigations.

The synthesis of silica-modified phenolic-impregnated chopped carbon fiber (C.F.) was accomplished through the process of sol-gel polymerization. The incorporation of silica into phenolic resin aerogels led to improved anti-oxidation properties, enhanced thermal stability, and a finer microstructure [26].

The synthesis of a composite material composed of carbon fabric impregnated with phenolic resin was achieved with the addition of ZrSiO_4_ sol as a fabric modifier. Resistance to ablation and shear strength between the laminae of the fabricated composite were tested and considered for evaluation. Zirconium dioxide (ZrO_2_) is formed during ablation, producing a protective film and reducing oxidation. The coating of ZrSiO_4_ sol improved the bonding between the reinforcement and matrix phases. Hence, improvements in interlaminar shear strength and ablation performance by 11% and 30%, respectively, were observed [27].

The carbon-fiber surface undergoes a modification process through facile impregnation with natural rubber latex. These modified carbon-fiber/phenolic-resin composites induce flexibility and lightweight characteristics. The study also investigated ablation, flexural, and thermal-insulation properties. After undergoing optimal modifications, the modified carbon-fiber/phenolic-resin composites demonstrated an increase in fracture strain and a reduction in flexural modulus. Additionally, they exhibited notable properties of ablation, thermal insulation, and low density [28]. These advancements herald a new era of carbon-phenolic ablatives. During re-entry to Earth, air-pyrolysis gas mixtures are produced, while CO_2_-pyrolysis gas mixes are generated during re-entry to Mars [29].

A model of transient heat conduction was introduced to assess the thermal-insulation properties of a passive thermal-protection system comprising layers of carbon-phenolic ablative material. This study investigates the pyrolysis characteristics of carbon-phenolic composites, specifically focusing on the gas components generated during pyrolysis and the resulting mass loss. The composite material undergoes pyrolysis at a temperature of 200 °C and exhibits a mass-loss rate of 25% when subjected to a temperature of 900 °C [30].

Carbon-and-phenolic composites were studied in a special micro-pyrolysis unit to identify and quantify products formed during pyrolysis. The highly sensitive detectors of the pyrolysis unit were operated between 300 °C and 800 °C. More than 50 different pyrolysis products were identified during the test [31].

A phenolic-impregnated carbon ablator (PICA) was created, and the process of PICA’s decomposition was examined under varying heating rates. The primary constituents influencing molar yields were carbon monoxide, methane, hydrogen, and water resulting from pyrolysis. The rate at which PICA’s phenolic resin decomposed was notably influenced by the heating rate [32].

Next, 0.3% graphene nanoplatelets (GNP) were incorporated into phenolic-matrix composites using carbon-fiber hand-layup techniques. Scanning electron microscopy (SEM) and impact, flexural, thermogravimetric, and ablation tests were conducted to analyze the improvement of thermal and mechanical properties. Comparisons were made with graphite powder-containing carbon-phenolic composites [33].

Mesoporous silica particles and carbon black filler were introduced into continuous carbon fiber/resol-type phenolic-resin composites. Tests were conducted to examine linear erosion rate, mass erosion rate, and insulation index considering different volumes and weight fractions of fillers [34].

Carbon-phenolic laminates with different lamina angles, i.e., 0° and 30°, and SiC-coated carbon-carbon composites were prepared. It was observed that the 30° carbon-phenolic composite had low abrupt material characteristics and lower internal temperatures, making it a more suitable candidate for thermal-protection systems [35].

Nanocomposites based on nano-boron carbide and phenolic material, as well as bulk molding compounds (BMC) incorporating carbon fibers, were formulated. The experimental results demonstrated that the carbon-fiber-based BMCs exhibited enhanced resistance to oxidation and increased thermal stability. Dimensional stability and structural integrity were also improved [36].

In the present investigation, hydroxyl-terminated polybutadiene (HTPB) and chopped carbon fibers were utilized as fillers within a matrix composed of novolac and resole phenolic resins. The impact strength, tensile strength, and heat resistance of the developed composites were assessed. Among the various compositions, the composite containing 2.2 wt.% of HTPB and 55.48 wt.% of carbon exhibited the highest tensile strength. Additionally, the combination consisting of 12.8 wt.% HTPB and 38.41 wt.% carbon exhibited the highest impact strength compared to other combinations. The study revealed that the composite material under consideration exhibited enhanced thermal resistance compared to pure phenolic resins [37].

In this study, carbon preforms were subjected to milling and used as a strengthening element, while phenolic resin served as the surrounding matrix. nanostructured phenolic-impregnated carbon ablator (n-PICA) was crafted through the incorporation of multi-walled carbon nanotubes (MWNTs) and nanoscale clays as additional materials. Results showed the formation of a low-density composite with improved dimensional stability and mechanical resistance [38].

X-ray microtomography was used to create three-dimensional representations of the internal structure of carbon-and-phenolic samples after ablation testing, and porosity and density were calculated based on microtomography [39].

Two different variants of 2.5D woven composites were produced, with carbon as the strengthening agent and phenolic resin with boron modifications as the binding matrix. These composites were designated as CDP (carbon/boron-modified phenolic 2.5D woven composites with 1.46 g/cm^3^ resin concentration) and CLP (carbon/boron-modified phenolic 2.5D woven composites with 1.23 g/cm^3^ resin concentration). The CDP had superior performance in terms of linear ablation resistance, but the CLP demonstrated a comparatively lower rate of mass ablation. The strength and modulus of CLP exceeded those of CDP [40].

In this study, a hybrid aerogel comprising silica and phenolic material served as the matrix, while a blend of quartz-needled carbon fibers was employed as the reinforcement component. The resulting composites showcased a remarkable balance of attributes, including low thermal conductivity and elevated compressive strength, with values varying between 5.96 and 17.01 MPa. The lightweight composite material exhibited favorable insulating and ablative properties [41].

The vacuum impregnation procedure was employed to fabricate an aerogel composite comprising needled carbon-fiber felt and phenolic resin (NCF-Ph). The Ph aerogels were utilized as fillers, while the NCF functioned as a matrix. The lightweight composites exhibited superior thermal protection and heat-insulation capabilities [42].

The siliconoxycarbide-phenolic interpenetrating aerogel matrix was impregnated with needle carbon fiber. The porosity and other attractive characteristics of siliconoxycarbide improved the composite’s elasticity, heat insulation, and compressive strength. The rate of linear ablation was greatly improved [43].

The incorporation of a small quantity of graphene oxide (GO) into the carbon-and-phenolic composite led to a significant improvement in its ablation resistance. GO fostered the development of graphitized crystals within the carbonized phenolic resin. This assertion was substantiated through diverse analytical methods, including scanning electron microscopy, X-ray diffraction, Raman spectroscopy, and transmission electron microscopy [44].

The ablative resistance of a carbon-and-phenolic ablator loaded with acidified graphitic carbon nitride (ag-C_3_N_4_) was evaluated. The addition of 0.2 wt.% of ag-C_3_N_4_ resulted in a noteworthy enhancement of the ablation properties, with a substantial increase of 69.23% [45].

Fibers treated with carbon nanotubes (CNTs) via the chemical vapor-deposition technique exhibited enhanced interfacial properties within carbon-fiber and phenolic-resin ablative materials. Moreover, an investigation examined the bending strength, impact strength, tensile strength, shear strength, and thermal conductivity after the modification process. A significant enhancement was observed through the process of alteration [46].

The synthesis of carbon-phenolic (C-Ph) nanocomposites was carried out using the compression-molding approach. In this process, several inorganic nanofillers, including nano zirconia, nano titania, and fumed silica, were used at varying loading percentages. The evaluation encompassed measurements of thermal erosion, thermal conductivity, and temperatures at the back wall. Additionally, scanning electron microscopy (SEM) was employed for analysis. Through these assessments and investigations, a more comprehensive understanding of the ablation, thermal, and mechanical attributes of carbon-phenolic (C-Ph) nanocomposites was achieved [47].

Nanoparticles such as zirconium diboride (ZrB_2_) and SiC are categorized as ultra-high-temperature ceramics because of their remarkable capacity to withstand ablation even in demanding high-temperature environments. By integrating nanoparticles as reinforcement within the carbon-and-phenolic material, a composite known as carbon/phenolic-ZrB_2_-SiC (C/Ph-ZS) nanocomposite is produced. Introducing ZrB_2_-SiC nanoparticles at a concentration of 7% into carbon/phenolic composites led to a notable 23% decrease in the linear ablation rate. Examination of the ablated surface of C/Ph-ZS nanocomposites revealed the creation of a uniform and compact layer comprising zirconium oxide (ZrO_2_) and silicon dioxide (SiO_2_), effectively mitigating oxidation [48].

Phenolic resin (Ph) modified with a ceramic precursor of silicon-containing polyborazine (SPBZ) was impregnated with carbon fiber and formed through hot compression. Laminated composites prepared had relatively higher inter-laminar shear stress and shape-retention abilities compared to C/Ph laminated composites. Also, they showed better ablation properties compared to C/Ph composites [49].

The heat capacity and pyrolysis heat of carbon/phenolic ablative materials were calculated using a model. The relative standard deviation for heat capacity was 10%, while that for the heat of pyrolysis was 20%. Furthermore, the model’s applicability extended to other composites based on carbon materials [50].

The vacuum-impregnation technique was employed to fabricate composites composed of carbon/phenolic (C/Ph) materials modified with zirconium carbide (ZrC). The research focused on exploring their thermal stability and ablation behavior through methods such as thermogravimetric analysis and plasma-wind-tunnel testing. Furthermore, the study encompassed experimental approaches like energy-dispersive X-ray spectroscopy, scanning electron microscopy, and X-ray diffraction to analyze the materials. Increasing the Zr content led to a rise in char yield while concurrently causing a decrease in back-face temperatures and linear ablation rates [51].

A composite material consisting of carbon-and-phenolic (C-Ph) with the addition of silicon carbide (SiC) was successfully fabricated. To assess the effects of the alteration, a series of experiments were done, including thermal-conductivity testing, compression tests, thermogravimetry analyses, plasma-wind-tunnel tests, and scanning electron microscopy. These experiments aimed to analyze the influence of the modification on mechanical, ablation, and thermal characteristics. The results indicate that C-Ph composites, when adjusted with a 5 wt.% SiC content, exhibited excellent characteristics [52].

The manufacture of carbon–phenolic composites modified by TaSi_2_/ZrSi_2_ involved compression molding techniques. The composites that underwent modification exhibited a high level of resistance to laser ablation. Furthermore, it was noted that prolonging the duration of laser ablation led to a reduction in the ablation rate [53].

Carbon-phenolic composites with ZrSi_2_ modifications were fabricated using compression-molding techniques. The objective of the current study was to assess the impact of various ZrSi_2_ weight percentages on thermal stability and ablation rate. The addition of ZrSi_2_ nanoparticles at a concentration of 5 wt.% led to increased linear and mass ablation rates in the composites [54].

Titanium diboride particles were incorporated to improve the heat stability of phenolic resins. The incorporation of TiB_2_ particles at a weight percentage of 20% into the phenolic-resin matrix resulted in a significant enhancement of the flexural strength, exhibiting a remarkable rise of 148% at a temperature of 1000 °C [55].

Organo-modified montmorillonite (o-MMT) nanoclay was reinforced into carbon–phenolic (C-Ph) composites, and mechanical, thermal, and ablation attributes were studied. The incorporation of nanoclay at a concentration of 2 wt.% resulted in enhancements in the flexural modulus, flexural strength, and interlaminar shear strength (ILSS), with these characteristics attaining optimal values [56].

Polycarbosilane (PCS) was introduced as an interface layer, while various concentrations of borosilicate glass were incorporated as fillers into carbon–phenolic (C-Ph) composites. The oxidation mechanism of composites changed with the addition of borosilicate glass. SiC was formed during the ablation process and exhibited a high melting point. With the increase in filler concentration, thermal conductivity was reduced. This effect lowered the back-face temperature of the composite, hence improving its high-heat-shielding performance [57]. 

A thermal-protection system was manufactured using carbon-and-phenolic material, and nano-alumina (Al_2_O_3_) particles were incorporated into the mixture. The altered system’s thermochemical properties were then examined. The results from the experiments suggested that the composites created displayed improved ablation performance [58].

The present study aimed to explore the linear and mass ablation rates of SiBCN–phenolic resin impregnated with carbon. The enhanced thermal stability of the modified phenolic resin was evaluated using thermal gravimetric analysis (TGA). The examination of oxidized products generated during ablation involved the utilization of Fourier transform infrared spectroscopy (FTIR), scanning electron microscopy (SEM), and X-ray diffraction (XRD) techniques. The findings of the study revealed a notable decrease in both the linear and mass ablation rates [59].

Modified composites composed of carbon fiber and phenolic resin were produced by incorporating modifications involving Si, borosilicate glass, and ZrSi_2_. Ablation and oxidation attributes were studied for the devised composites. The outcomes demonstrated enhanced thermal insulation and ablation resistance [60].

SiC, ZrB_2_, and glass microspheres modified with phenol-formaldehyde resins were reinforced with carbon-fiber fabric. The proposed composite was thermally stable up to 422 °C, and the strength was reduced by 50% after plasma-stream exposure [61].

In the presented work, carbon nanotubes and silicon carbide nanoparticles were introduced into carbon-and-phenolic ablators. The effect of nanoparticles was checked for ablation performance. Results revealed that carbon nanotubes had a more positive impact on ablation performance than did silicon carbide [62].

Two carbon-and-phenolic material composites were formulated, incorporating the inclusion of graphene oxide (GO) and graphitic carbon nitride (g-C_3_N_4_) for modification. The composites treated with GO and g-C_3_N_4_ showed a significant enhancement in ablation resistance, with improvements of 62.02% and 22.36%, respectively [63].

To explore the decomposition and gradual surface erosion of phenolic-impregnated carbon ablative material, a finite-element-analysis (FEA) model was used. Using predefined surface conditions and gas compositions, the model allowed the calculation of recovery enthalpy and the convective heat-transfer coefficient [64].

To learn more about the ablative properties of carbon-and-phenolic composites, researchers have used both experimental investigation and mathematical simulation models. The test was conducted under the influence of a plasma jet. Specific-mass-loss rates, mass-loss rates, and back-surface temperature were studied with plasma-jet-exposure time. Results exhibited that the results of the simulation model accorded well with experimental results [65].

Two distinct carbon–phenolic ablative materials, both with a density of 0.3 g/cm^3^, were synthesized. The thermal properties of these fabricated ablators were subsequently evaluated in a hypersonic plasma-wind-tunnel facility. Also, a mathematical model was prepared based on a finite element-based model. Results showed a good match between experimental and theoretical values [66].

This study focused on modeling the thermal behavior and surface recession of a carbon-and-phenolic ablative material, which possesses notable porosity and a reduced density [67].

A pyrolytic model employing a multi-objective genetic algorithm was developed to investigate the characteristics of phenolic-impregnated carbon ablative material. The model was designed to operate under high rates of heating, and its calibration was achieved by utilizing precise data obtained from actual thermal decompositions. The present model incorporates the quantification of mass loss and the characterization of gases generated throughout the process of pyrolysis [68].
polymers-16-01461-t001_Table 1Table 1Ablation Characteristics of Phenolic-Based Ablative Composites.Phenolic Ablative Composites in LiteratureAblation CharacteristicsMethod of Ablation TestReferences% Reduction in LAR% Reduction in MARNeedled felt carbon- quartz fiber/phenolic-silica hybrid aerogel (C-QF/Ph-Si75)84.481Oxy-acetylene torch test at 2000 °C for 300 s[41]7.5 wt.% of ZrSi_2_ + carbon–phenolic69.646.3Oxy-acetylene torch test at a heat flux of 4.28 × 106 W/m2 for 30 s[54]Acidified graphitic carbon nitride-carbon/phenolic 0.2 wt.% ag-C_3_N_4_-CF/Ph6927Oxy-acetylene torch test at ~2900 °C[45]0.1 wt.% GO + carbon and phenolic62-Oxy-acetylene torch test at 3000 °C for 30 s[63]15 wt.% borosilicate glass + polycarbosilane (PCS) + C-Ph628.5Oxy-acetylene torch test for 60 s[57]5 wt.% SiC + C-Ph60-plasma windtunnel, heat flux of 1.6 × 106 W/m^2^ for 50 s[52]5 wt.% silicon carbide + 0.1 wt.% MWCNT+carbon fibre and phenolic resin43-Oxy-acetylene torch test for 20 s[62]TaSi_2_/ZrSi_2_ + carbon–phenolic2943laser ablation, heat flux 1 × 107 W/m^2^ for irradiation of 30–100 s[53]SiBCN–phenolic-C2717Oxy-acetylene torch test for 30 s[59]7 wt.% ZrB_2_-SiC + carbon and phenolic (C-Ph-ZS7)23-Oxy-acetylene torch test at 2500 °C for 160 s[48]ZrSiO_4_ sol + carbon fabric + phenolic resin-21Oxy-acetylene torch test for 30 s[27]6 wt.% organo-modified montmorillonite (o-MMT) nano clay + C-Ph-35Oxy-acetylene torch test heat flux 5 × 106 W/m^2^[56]


## 3. Carbon–Elastomeric Ablative Composites

Carbon, as a strengthening component, is widely accessible in various forms, including as continuous fibers, chopped fibers, fiber fabrics, carbon-fiber prepreg, and particulates, among others. These components can be produced with different manufacturing techniques, which were already discussed earlier [22,23,24,25].

Some elastomeric matrices commonly used in ablatives are polyurethane, silicone-based elastomers, natural rubber, etc. Polyurethane has high flexibility, toughness, and resistance to erosion, while silicone-based elastomers show resistance to oxidation and good thermal stability when various fillers and carbon are introduced as reinforcement [20,21]. 

The significance of filler in ablatives was already discussed in an earlier section. Fillers such as CeO_2_, SiC, BER(4,4-bis(3-(oxiran-2-ylmethoxy) benzyl)-1,1′-biphenyl), titanium dioxide (TiO_2_), etc., were used to optimize the characteristics of carbon-and-elastomeric ablative composites. Tensile strength was improved with BER modifications in carbon and elastomeric ablatives [69]. CeO_2_ enhanced the thermal stability and mechanical strength of a phenyl silicone-rubber-based composite [70]. Elongation at break, hardness, and tensile strength were increased by SiC modification [71]. As with carbon-and-phenolic ablative composites, a table (Table 2) is introduced below, which shows a % reduction in both LAR and MAR, as determined by various authors. These data provide insight into the design of ablative composites. So, recent pathways based on carbon and elastomeric ablative composites are reported here. 

Silicone-rubber composites modified with BER were fabricated in a sequence involving melting, mixing, and cold pressing. The inclusion of BER led to heightened strength in the ceramic layer and the development of well-structured graphitic carbon formations. Additionally, the overall tensile strength of the composite was enhanced [69]. CeO_2_- and graphene-modified phenyl silicone-rubber composites were studied. According to the findings, even a very small quantity of fillers improved the material’s mechanical strength and thermal stability [70]. 

The introduction of silicon carbide (SiC) and carbon fiber (C.F.) at different parts per hundred parts of resin (phr) was performed in ethylene–propylene–diene-based composites. The analysis and results revealed that composites containing C.F. at 10 phr and SiC up to 20 phr showcased enhanced attributes, including modulus, hardness, tensile strength, and elongation at break (725%) [71].

The effect of different carbon-fiber lengths was investigated in ablative composites made from silicone rubber with incorporated ceramic fillers. According to the research findings, increasing the fiber length from 0.5 to 3 mm led to a drop in the linear ablation rate of 0.144 mm per second. Furthermore, the back-face temperature decreased from 117.7 to 107.9 °C. Additionally, enhanced thermal-insulation performance was achieved due to the combination of low thermal conductivity and the formation of a substantial ceramic layer [72].

Carbon fibers (C.F.), quartz fibers (Q.F.), aramid fibers (A.F.), and poly (p-phenylene benzobisoxazole) fibers (PBO) were the four different types of reinforcements utilized in conjunction with an elastomeric matrix composed of epoxy-resin-modified liquid silicone rubber. Comparative analyses were conducted to assess the thermal-insulation attributes and linear ablation rates of the composites. The findings revealed that composites containing quartz fibers (Q.F.) exhibited the highest ablation rates [73].

Carbon fibers and magnesium carbonate (MgCO_3_)-added silicone-rubber composites were proposed to examine the ablation-resistance properties. The charred structure of composites containing 10 parts per hundred parts of resin (phr) of MgCO_3_ demonstrated both low thermal conductivity and satisfactory strength. Compared to the unaltered base material, the incorporation of MgCO_3_ along with carbon fibers resulted in a decrease in the rate of linear ablation of the composites by 30.76% [74].

The evolution of thermophysical properties was observed with the introduction of silicon carbide powder (SCP) and carbon fibers in silicone-based composites. The experimental data were used as a basis for the calculations of thermophysical parameters such as specific heat capacity, density, and thermal conductivity. These experimentally determined properties are utilized in a 1D SAMCEFTM finite-element code. The outcomes of the analytical framework exhibited the thermal efficiency of the materials, confirming that composites could be used as thermal-protection systems (TPS) [75].

Mixing modest amounts of carbon nanotubes (CNTs) with alumina (Al_2_O_3_) powder increased the methyl vinyl silicone-rubber matrix composite’s hardness, thermal conductivity, and Young’s modulus. Results showed that hybridized fillers improved matrix–filler interaction [76].

Polyarylacetylene (P.A.) filler was incorporated into silicone-rubber–carbon woven laminates (SRWL), and its impact was examined. The outcomes revealed that the peak back-face temperature reached 71 °C and that the inclusion of 10 parts per hundred parts of resin (phr) of P.A. led to improved ablation properties in the modified SRWL [77].

Boron phenolic-resin inorganic nanofibers, multi-walled carbon nanotubes (MWCNTs), SiC nanowires, and methyl-polyhedral oligomeric silsesquioxanes (methyl–POSS) with ethylene propylene diene monomer (EPDM) composites were proposed to examine ablation characteristics. Results showed that CNTs and SiC nanowires reduced the porosity of the charred layer, while boron phenolic resin had a relatively higher char yield. Methyl–POSS exhibited the best ablation performance and the lowest mass ablation rate [78].

Silicone-rubber-based composites were prepared with different reinforcements such as glass, carbon, ceramics, and silica fiber. Results showed that glass-fiber-reinforced composites had better insulating properties [79].

Graphene, modified fumed silica (MFS), and titanium dioxide (TiO_2_) were employed as additives in composites composed of ethylene propylene diene monomer (EPDM) and silicone rubber. This study investigated the dielectric and thermal properties of the newly created composites. The results demonstrated that the EPDM rubber composites exhibited improved thermal conductivity (approximately 7–35%) and enhanced thermal stability (approximately 30–50 °C) [80].

Blending aramid fiber (A.F.) with carbon fiber introduced flexibility, elevated thermal stability, and reduced thermal conductivity to composites based on ethylene propylene diene monomer (EPDM). This hybridization led to a 31% enhancement in thermal efficiency compared to composites made solely of A.F.–EPDM [13].

Carbon nanotubes (CNTs) facilitate the vapor deposition of pyrolysis gases onto the charred layer. This distinctive property renders CNTs a valuable filler in ethylene propylene diene monomer (EPDM) composites. Analysis indicated that the thermal conductivity and ablation rates of the charred layer’s framework in CNT-filled EPDM composites were lowered [81].

Carbon nanotubes (CNTs)–filled ethylene–propylene diene composites were investigated for ablation rate in a particle-erosion environment. CNT-reinforced composites exhibited superior thermal resistance. The inclusion of CNTs led to a 68% reduction in the charring rate compared to composites without CNTs [82].

Composites based on ethylene propylene diene monomer (EPDM) were formulated, incorporating carbon-fiber fabric, asbestos, and Vulkasil-C fillers. Incorporating Vulkasil-C enhanced the ablative performance and mechanical properties of EPDM-based composites, concurrently reducing the swelling index compared to asbestos and carbon-fiber fabric [83]. Similarly, the introduction of carbon nanofibers (CNFs) into polyisoprene elastomer (P.R.) composites elevated ablation resistance and decreased the back-face temperature [84].

Grainy (nano-diamonds), layered (graphene nanoplates), and fibrous (nanotubes) materials were reinforced with polymerized styrene–butadiene rubber (S–SBR) composites. 2% of nano-diamonds by volume doubled the rupture strength of S–SBR-based composites [85]. Carbon black (C.B.)- and carbon nanotube (CNT)-filled fluoroelastomer (F.E.) ablatives were prepared. Carbon black (C.B.) and carbon nanotube (CNT) surfaces were modified with acidic and basic media. TGA-GCMS and TGA analysis were performed for the prepared composites. Results exhibited that surface-modified nano-fillers produced greater quantities of CF2–CF2 products during ablation, which increased the thermal stability of F.E. [86].

A small amount of multilayer graphene (MLG) nanoparticles was suggested to replace the high inclusion of aluminum trihydroxide (ATH) and carbon black (C.B.) fillers in polybutadiene/chloroprene (BR/CR) nanocomposites without altering their useful properties [87].

Waste leaves filled with carbon were used to prepare carbon and polyurethane composites. The inclusion of different weight percentages of carbon fibers (C.F.) was investigated. Both linear and mass ablation rates were found to be reduced with increased C.F. content. XRD and SEM analysis confirmed that nanocomposites were formed due to heat treatment at 900 °C [88].
polymers-16-01461-t002_Table 2Table 2Ablation Characteristics of Phenolic-Based Ablative Composites.Elastomeric Ablative Composites in LiteratureAblation CharacteristicsMethod of Ablation TestReference% Reduction in LAR% Reduction in MARWaste leave side filled with carbon fiber + polymeric methylene diphenylene diisocyanate (PMDI)8360Oxy-acetylene torch test at 2400 °C for 30 s[88]10 phr Polyarylacetylene (P.A.) + silicone rubber-carbon woven laminates (SRWL)71.60-Oxy-acetylene torch test[77]20 wt.% BER(4,4-bis(3-(oxirane-2-ylmethoxy)benzyl)-1,1′-biphenyl) + silicone rubber5430Oxy-acetylene flame test at a heat flux of 4.152 × 106 W/m^2^ for 30 s[69]3 mm carbon fibers + ceramic filled silicone rubber53.664.4Oxy-acetylene torch test at a heat flux of 4.57 × 106 W/m^2^ for 30 s[72]Aramid fiber (AF) + carbon fiber + ethylene propylene diene monomer (EPDM)31-Oxy-acetylene flame test at a heat flux of 4.57 × 106 W/m^2^ for 30 s[13]10 phr Magnesium carbonate (MgCO3) + carbon fibers (CFs) + silicone rubber30.76-Oxy-acetylene torch test for 30 s[74]methyl–polyhedral oligomeric silsesquioxanes (methyl–POSS) + ethylene propylene diene monomer (EPDM)-28-[78]CNTs + ethylene propylene diene monomer (EPDM)-10.7-[81]


## 4. Conclusions

An extensive literature review on the recent development of carbon-and-phenolic, modified-carbon-and-phenolic, carbon-and-elastomeric, and modified-carbon-and-elastomeric ablative composites was conducted. It was observed that different categories of fillers, like organic, inorganic, and ceramic fillers, were used for modification of both carbon-and-phenolic and carbon-and-elastomeric ablative composites, and they were studied vis-à-vis different characteristics such as ablation rate, thermal stability, erosion resistance, flexural strength, compressive strength, tensile strength, shear strength, fiber–matrix interaction, char formation, and thermal conductivity of ablative composites. The ablation characteristics of phenolic-and-elastomeric matrix-based ablatives in the reported works were studied. A reduction in both LAR and MAR determined by different authors was discussed; these characteristics play a critical role in selecting reinforcements, matrices, and fillers for TPS. It was observed that most of the reported work was based on experimental analysis, while a very limited amount of work was based on mathematical modeling. To understand the physics of ablation, it is essential to develop a mathematical model and a virtual analysis before the actual development of the physical model. This will help in finding an optimized composition of ‘reinforcements, matrices, and modifiers’ and would reduce product-development time, cost, and waste. 

## Data Availability

The original contributions presented in the study are included in the article, further inquiries can be directed to the corresponding author.

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
