# Peer review of "State-of-the-Art on Advancements in Carbon–Phenolic and Carbon–Elastomeric Ablatives"

_polymers, 2024, doi:10.3390/polym16111461_

Round 1

Reviewer 1 Report

Comments and Suggestions for Authors

In this paper, the author reviews ablation composites, especially Carbon/phenoic and Carbon Elastomeric ablatives. It is worth reading as a review because it summarizes and presents a number of papers. However, there are a few problems. Therefore, I labeled this revision as major. Detailed comments are provided below:

i)                   Tables 1 and 2 show the LAR and MAR. What order are these in? It is easier to understand if there is a law.

ii)                 Why do you focus on MAR and LAR? The author presents the results of various papers from line 115, but does not mention.

iii)               The paragraph was never changed from page 3, line 115 to page 7, line 323. It is just a list of information and very difficult to read. Can't you separate the paragraphs a little more, or insert more figures, etc.?

iv)                Table 1 shows that there are different methods of ablation testing, but does it make sense to compare those results in Table 1?

Author Response

The authors sincerely acknowledge the comments and suggestions of the reviewers who have been instrumental in improving and upgrading the paper in its present form.

REVIEWER #1

COMMENT #1

In this paper, the author reviews ablation composites, especially Carbon/phenoic and Carbon Elastomeric ablatives. It is worth reading as a review because it summarizes and presents a number of papers. However, there are a few problems. Therefore, I labeled this revision as major. Detailed comments are provided below:

# RESPONSE

The authors thank the reviewer for their appreciation.

COMMENT #2

Tables 1 and 2 show the LAR and MAR. What order are these in? It is easier to understand if there is a law.

# RESPONSE

There is no such law but the theory and implication of LAR and MAR have been discussed on Page 3 / 17 Lines nos. 110 – 114 wherein the importance and effect of these in connection to Ablative properties & characteristics are depicted. 

COMMENT #3

Why do you focus on MAR and LAR? The author presents the results of various papers from line 115 but does not mention.

# RESPONSE

The Ablation characteristics are quantified with the extent of reduction in LAR and MAR so, there are various methods of manufacturing and testing but to compare the same quantitatively the above parameters are focused.

Various results from different papers are indeed cited in the referred section including LAR & MAR for example while discussing reference [30] MAR has been depicted.

COMMENT #4

The paragraph was never changed from page 3, line 115 to page 7, line 323. It is just a list of information and very difficult to read. Can't you separate the paragraphs a little more, or insert more figures, etc.?

# RESPONSE

The authors are sorry for the inconvenience caused. The section in radar has been rewritten in the revised manuscript with each paragraph designated for each of the cited references.

COMMENT #5

Table 1 shows that there are different methods of ablation testing, but does it make sense to compare those results in Table 1?

# RESPONSE

The authors have tried to enlist different testing methods but surely the ultimate requirement is to reduce the material loss hence, the values of LAR and MAR are provided so that researchers would get an idea of the selection of testing method for a desirable LAR and / MAR

Reviewer 2 Report

Comments and Suggestions for Authors

It is not possible to read the lables at Graphical Abstract.

Please, heat up this statement by literature: Carbon fibers (C.F.), quartz fibers (Q.F.), aramid fibers (A.F.), and poly (p-phenylene benzobisoxazole) fibers (PBO) were the four different types of reinforcements that were utilized in conjunction with an elastomeric matrix composed of epoxy resin-modified liquid silicone rubber [DOI: 10.1088/1402-4896/ad070c;…etc].

Graphical representation of the manuscript is very pure. I suggest providing summarization in graphs and schemes.

The conclusions should be given to every paragraph.

I suggest authors to come through the rules for review writing.

Comments on the Quality of English Language

The language should be improved. There are many typos and the style is not suitable.

Author Response

The authors sincerely acknowledge the comments and suggestions of the reviewers who have been instrumental in improving and upgrading the paper in its present form.

REVIEWER #2

COMMENT #1

It is not possible to read the labels at Graphical Abstract.

# RESPONSE

The authors agree that the labels in the Graphical Abstract are not legible. The authors just wanted to compare the trend of work on the two types of ablative composites for which this review paper has been drafted. Moreover, the figures have been provided with 500 dpi resolution and hence labels can be very clearly seen on zooming.

COMMENT #2

Please, heat up this statement by literature: Carbon fibers (C.F.), quartz fibers (Q.F.), aramid fibers (A.F.), and poly (p-phenylene benzobisoxazole) fibers (PBO) were the four different types of reinforcements that were utilized in conjunction with an elastomeric matrix composed of epoxy resin-modified liquid silicone rubber [DOI: 10.1088/1402-4896/ad070c;…etc].

# RESPONSE

The reference [73], provided at the end of the paragraph, is the reference which supports the statement

COMMENT #3

Graphical representation of the manuscript is very poor. I suggest providing summarization in graphs and schemes.

# RESPONSE

The authors understand that generally, a graphical representation provides a better visual understanding but, in this case, as it is a review work, the authors felt that a detailed explanation would be a better tool for the reader to understand and use the gained knowledge to carry out their research.

COMMENT #4

The conclusions should be given to every paragraph.

# RESPONSE

Each paragraph is the conclusion of each cited work so a conclusion in the end would be a better option, the author feels.

COMMENT #5

I suggest authors to come through the rules for review writing.

# RESPONSE

The authors thank the reviewer for the suggestion and will try to follow the same from now onwards and will try to incorporate the suggestions provided in the present paper in all forthcoming review papers.

COMMENT #6

The language should be improved. There are many typos and the style is not suitable.

# RESPONSE

The Complete manuscript has been revised to eradicate the typos and elevate the language. 

Reviewer 3 Report

Comments and Suggestions for Authors

 This review article analyses recent advances in carbon/phenolic and carbon/elastomer composites. Various factors associated with ablative composites are considered, including their ablation erosion rate, their ability to withstand high temperatures, their tensile, flexural and compressive strengths, how well the fibres and composite matrix interact, and character creation during the process. Calculations by different authors of the percentage reduction in linear ablation rate (LAR) and mass ablation rate (MAR) are discussed. The study of these properties determines promising trajectories in the field of carbon-phenolic and carbon-elastomer ablation composites.

1. It should be added that carbon nanotubes with different morphologies have different effects on the properties of elastomeric composites: 

Jia, S.L.; Geng, H.Z.; Wang, L.; Tian, Y.; Xu, C.X.; Shi, P.P.; Gu, Z.Z.; Yuan, X.S.; Jing, L.C.; Guo, Z.Y.; et al. Carbon nanotube-based flexible electrothermal film heaters with a high heating rate. R. Soc. Open Sci. 2018, 5, 172072. 

Aouraghe, M. A.; Xu, F.; Liu, X.; Qiu, Y. Flexible, rapidly responsive and highly efficient E-heating carbon nanotube film. Compos. Sci. Technol. 2019, 183, 107824.

https://doi.org/10.3390/polym15010249

https://doi.org/10.3390/polym16060774.

2. In Section 3. Carbon-Elastomeric Ablative Composites, information on the parameters of the carbon nanotubes used should be entered and their parameters should be tabulated.

3. Conclusions are poorly formatted. Numerical parameters should be included.

Comments on the Quality of English Language

Minor editing of English language required

Author Response

The authors sincerely acknowledge the comments and suggestions of the reviewers that have been instrumental for improving and upgrading the paper in its present form.

REVIEWER #3

COMMENT #1

This review article analyses recent advances in carbon/phenolic and carbon/elastomer composites. Various factors associated with ablative composites are considered, including their ablation erosion rate, their ability to withstand high temperatures, their tensile, flexural and compressive strengths, how well the fibres and composite matrix interact, and character creation during the process. Calculations by different authors of the percentage reduction in linear ablation rate (LAR) and mass ablation rate (MAR) are discussed. The study of these properties determines promising trajectories in the field of carbon-phenolic and carbon-elastomer ablation composites.

# RESPONSE

The authors thank the reviewer for their appreciation.

COMMENT #2

It should be added that carbon nanotubes with different morphologies have different effects on the properties of elastomeric composites: 

Jia, S.L.; Geng, H.Z.; Wang, L.; Tian, Y.; Xu, C.X.; Shi, P.P.; Gu, Z.Z.; Yuan, X.S.; Jing, L.C.; Guo, Z.Y.; et al. Carbon nanotube-based flexible electrothermal film heaters with a high heating rate. R. Soc. Open Sci. 2018, 5, 172072. 

Aouraghe, M. A.; Xu, F.; Liu, X.; Qiu, Y. Flexible, rapidly responsive and highly efficient E-heating carbon nanotube film. Compos. Sci. Technol. 2019, 183, 107824.

https://doi.org/10.3390/polym15010249

https://doi.org/10.3390/polym16060774.

# RESPONSE

The authors agree with the suggestion of the reviewer but would like to state that in the present work, the emphasis is on the phenolic and elastomeric matrix with carbon reinforcement where focus and variation have been on matrix and not the forms of reinforcement. The reinforcement has been applied but based on the class i.e. Continuous Fibre, Chopped Fibre, Particulate Fillers (Macro-, Micro, Nano- etc.).

The suggested domain is a very lucrative one and it can be taken exclusively as a separate work.

COMMENT #3

In Section 3. Carbon-Elastomeric Ablative Composites, information on the parameters of the carbon nanotubes used should be entered and their parameters should be tabulated.

# RESPONSE

The authors acknowledge the suggestion and will try to put up the same in their next assignment.

COMMENT #4

Conclusions are poorly formatted. Numerical parameters should be included.

# RESPONSE

The authors are of the view that Conclusions are generalizations of results and generally numerical parameters (which are parts of Results & Discussion) should be avoided if not very essential. But still, if the reviewer/editor wants it can be modified accordingly.

COMMENT #5

Minor editing of English language required

# RESPONSE

The Complete manuscript has been revised to eradicate the typos and elevate the language. 

Round 2

Reviewer 2 Report

Comments and Suggestions for Authors

The manuscript has not been revized sufficiently.

Comments on the Quality of English Language

English should be improved.

Author Response

COMMENT #1

The manuscript has not been revised sufficiently.

# RESPONSE

The complete manuscript has been carefully and thoroughly revised to correct all typographical and grammatical errors. All changes are highlighted with yellow.

COMMENT #2

English should be improved.

# RESPONSE

The Complete manuscript has been carefully and thoroughly revised to eradicate the typos and elevate the standard of English. All changes are highlighted with yellow.

Reviewer 3 Report

Comments and Suggestions for Authors

Accept in present form

Author Response

Thank you for your encouraging words. 

Round 3

Reviewer 2 Report

Comments and Suggestions for Authors

The paper can be published.